# An Automated Functional Annotation Pipeline That Rapidly Prioritizes Clinically Relevant Genes for Autism Spectrum Disorder

**DOI:** 10.3390/ijms21239029

**Published:** 2020-11-27

**Authors:** Olivia J. Veatch, Merlin G. Butler, Sarah H. Elsea, Beth A. Malow, James S. Sutcliffe, Jason H. Moore

**Affiliations:** 1Department of Psychiatry and Behavioral Sciences, University of Kansas Medical Center, Kansas City, MO 66160, USA; mbutler4@kumc.edu; 2Department of Molecular and Human Genetics, Baylor College of Medicine, Houston, TX 77030, USA; Sarah.Elsea@bcm.edu; 3Sleep Disorders Division, Department of Neurology, Vanderbilt University Medical Center, Nashville, TN 37232, USA; beth.malow@vumc.org; 4Vanderbilt Genetics Institute, Department of Molecular Physiology & Biophysics, Department of Psychiatry and Behavioral Sciences, Vanderbilt University School of Medicine, Nashville, TN 37232, USA; james.s.sutcliffe@Vanderbilt.Edu; 5Department of Biostatistics, Epidemiology, & Informatics, University of Pennsylvania, Philadelphia, PA 19104, USA; jhmoore@upenn.edu

**Keywords:** bioinformatics, human genetics, pharmacogenomics, autism

## Abstract

Human genetic studies have implicated more than a hundred genes in Autism Spectrum Disorder (ASD). Understanding how variation in implicated genes influence expression of co-occurring conditions and drug response can inform more effective, personalized approaches for treatment of individuals with ASD. Rapidly translating this information into the clinic requires efficient algorithms to sort through the myriad of genes implicated by rare gene-damaging single nucleotide and copy number variants, and common variation detected in genome-wide association studies (GWAS). To pinpoint genes that are more likely to have clinically relevant variants, we developed a functional annotation pipeline. We defined clinical relevance in this project as any ASD associated gene with evidence indicating a patient may have a complex, co-occurring condition that requires direct intervention (e.g., sleep and gastrointestinal disturbances, attention deficit hyperactivity, anxiety, seizures, depression), or is relevant to drug development and/or approaches to maximizing efficacy and minimizing adverse events (i.e., pharmacogenomics). Starting with a list of all candidate genes implicated in all manifestations of ASD (i.e., idiopathic and syndromic), this pipeline uses databases that represent multiple lines of evidence to identify genes: (1) expressed in the human brain, (2) involved in ASD-relevant biological processes and resulting in analogous phenotypes in mice, (3) whose products are targeted by approved pharmaceutical compounds or possessing pharmacogenetic variation and (4) whose products directly interact with those of genes with variants recommended to be tested for by the American College of Medical Genetics (ACMG). Compared with 1000 gene sets, each with a random selection of human protein coding genes, more genes in the ASD set were annotated for each category evaluated (*p* ≤ 1.99 × 10^−2^). Of the 956 ASD-implicated genes in the full set, 18 were flagged based on evidence in all categories. Fewer genes from randomly drawn sets were annotated in all categories (x = 8.02, sd = 2.56, *p* = 7.75 × 10^−4^). Notably, none of the prioritized genes are represented among the 59 genes compiled by the ACMG, and 78% had a pathogenic or likely pathogenic variant in ClinVar. Results from this work should rapidly prioritize potentially actionable results from genetic studies and, in turn, inform future work toward clinical decision support for personalized care based on genetic testing.

## 1. Introduction

Autism spectrum disorder (ASD) is a heterogeneous neurodevelopmental condition characterized by impairments in social interactions, delays in language development and patterns of restricted interests and/or repetitive behaviors [1]. ASD manifests along a wide distribution of core symptom severity and numerous health conditions present with ASD [2,3]. It is well-established that ASD has a predominantly genetic etiology, and genetic factors influence many of the medical conditions diagnosed with ASD [4,5]. Advancements in genomic technology are continually generating large amounts of data implicating various biological processes dysregulated in ASD. Notably, ASD is one of the most complex and prevalent neurodevelopmental conditions observed in humans, with a worldwide prevalence estimated at one in 160 children [6]. Furthermore, insufficient evidence exists substantiating efficacy of pharmaceutical treatment of specific symptoms and co-occurring conditions in ASD with many reported adverse events [7,8,9]. Informing more effective, personalized approaches for treatment of these heterogeneous conditions is a necessary area of research. An important next step is to better understand how current data can inform diagnosis and treatment of co-occurring conditions in ASD [10].

Fortunately, excellent tools are available to help circumvent challenges related to interpreting results from human genetic studies and offer immediate insight into mechanisms that are most likely to translate in the clinic setting [11,12,13,14,15,16,17,18,19]. While the definition of what constitutes ‘clinically actionable’ genetic information can vary and is dynamic [10], it is defined in this project as any functionally relevant, disease-implicated gene with evidence indicating a patient may have co-occurring condition that requires direct intervention, or that variation in the gene (or genetic mechanism) may influence how a patient will respond to a drug.

To initially identify all of the potential genetic factors that contribute to expression of a complex condition like ASD, it is necessary to compile results from many sources. These include statistical associations (e.g., genome-wide association study (GWAS) hits, rare variant burden test results, transmission disequilibrium test results) and functional evidence in humans and model organisms. Centralized repositories, like DisGeNET (https://www.disgenet.org/), integrate and uniformly annotate data and allow easy access to comprehensive knowledge of the genetic underpinnings of disease [17].

One way to facilitate interpretation of genetic results is to focus on implicated genes expressed in tissues relevant to the disease etiology [13,20,21,22,23]. As most evidence indicates that ASD relates to neurodevelopment [24], identifying ASD-associated genes that are expressed at appreciable levels in human brain tissue can rapidly prioritize genes that are more likely to be functionally relevant. Integrated databases, such as the Expression Atlas (https://www.ebi.ac.uk/gxa/home), offer results from large microarray and RNA-sequencing studies in humans (e.g., the Genotype Tissue Expression (GTEx) Project) and allow for efficient mining of data related to tissue-specific gene expression [13,22].

To further annotate those genes that are more likely to be functionally related to ASD etiology, it is useful to focus on genetic mechanisms evidenced to underlie ASD symptom expression. Notably, the biological functions of different genes implicated in ASD point to convergent mechanisms [25,26,27]. By identifying biological processes enriched for genes implicated in ASD and then focusing specifically on genes that influence these processes, genes of particular interest for functional follow-up and drug target repurposing or discovery can be readily classified [28,29,30]. Efforts to consistently describe gene products across databases, like the Gene Ontology (GO) Consortium [16], allow for identification of these larger, multi-gene processes. Furthermore, transgenic techniques in model organisms are incredibly useful for identifying genes with functional consequences relevant to human disease [31,32,33] and drug discovery [34]. On-going efforts by groups like the International Mouse Phenotyping Consortium (IMPC) offer web portals which allow for rapid mining of mouse phenotype data from knockout mutant strains for eventually every protein-coding gene in mice [35]. Determining the traits that are associated with knocking out specific genes may also offer insight into which genes are more likely to influence expression of co-occurring conditions in ASD.

The ultimate goal of precision medicine is to prevent disease; however, realization of this vision is likely many years away [36]. A key opportunity in precision medicine is to incorporate drug ontology and pharmacogenomics data to inform care and tailor treatment (i.e., maximize efficacy, minimize adverse events) [37,38]. An approach toward this goal is to determine if any of the proteins encoded by ASD candidate genes are targets for pharmaceutical compounds that are currently approved to treat symptoms and co-occurring conditions in ASD, or are potential novel targets based on evidence that they are chemically similar or function in the same mechanisms as approved drug targets [39,40]; integrated knowledge bases, like Pharos (https://pharos.nih.gov/), exist to identify these proteins [14]. It is also of interest to know whether a patient with a given variant will respond differently to medications. Resources such as the Pharmacogenomics Knowledge Base (PharmGKB; https://www.pharmgkb.org/) coalesce these data into a common portal [41].

To prioritize genes that may be clinically relevant, specifically with regard to co-occurring conditions, but have yet to be confirmed as having a causal relationship with human disease (i.e., pathogenic), it is advantageous to incorporate evidence related to genes with known pathogenic variants. Resources that define the clinical relevance of variants, like the American College of Medical Genetics (ACMG) [42] and ClinVar [18], can be used to identify these ‘clinically actionable’ genes. Furthermore, protein-protein interaction databases, like STRINGdb (https://string-db.org/), can be used to identify different genes with disparate variants that impact a common network [43]. By extension, protein interaction knowledge may identify candidate genes more likely to house pathogenic variants via revealing direct connections between their protein products and the products of genes with known pathogenic variants.

A major goal of our study is to develop an automated bioinformatics pipeline using the databases referenced above that can identify genes pulled from all published studies of ASD where variation may indicate a patient has a co-occurring condition that is important to treat, or are more likely to be useful for drug development and/or pharmacogenomics approaches to treatment of symptoms and comorbidities in patients with ASD. Each step of the pipeline is evaluated by comparing results for genes cited in connection with ASD to results for 1000 sets of *n* = 956 randomly selected human protein coding genes of equal number to the ASD gene set. Prioritized ASD genes are then queried against a gold standard, defined as expert curated evidence in ClinVar indicating that the gene has a pathogenic variant.

## 2. Results

At the time of these analyses, there were 956 unique protein coding genes with evidence for associations with ASD in the DisGeNET, which reflects expansive evidence from candidate gene studies, genome-wide genotyping, whole-exome/genome sequencing, and functional studies in human cell lines and model organisms (Appendix A). Notably, this initial list of ASD candidate genes included genes implicated via evidence from genetic studies of idiopathic ASD cases (e.g., *NLGN1*, *NLGN2*, *NLGN3*, *NLGN4X* and *NLGN4Y*), as well as genes pulled from studies evaluating syndromic cases of ASD (e.g., *FMR1*, *MECP2*, and *SHANK3*). All of the functional attributes used in our approach for prioritizing candidate genes had more ASD-related genes when compared to 1000 random sets, each with a random selection of 956 genes (Table 1). Ultimately, 18 ASD candidate genes were prioritized based on annotation in each step of the pipeline, which was more than the average number of genes from random gene sets (Χ^2^ = 11.30, *p* = 7.75 × 10^−4^; Table 1).

### 2.1. ASD Candidate Gene Expression in Human Brain

There were 861 (90.1%) ASD genes which were expressed at TPM ≥ 0.5 in at least one of the brain regions evaluated using available RNA-sequencing data from typical human tissue. This was increased when compared to random gene sets (84.6%, Χ^2^ = 21.36, p_corrected_ = 5.07 × 10^−6^; Table 1). Notably, ASD-related genes were expressed at different levels in human brain tissue available in GTEx when compared to expression patterns of the random gene sets (*F*_(13, 13117)_ = 2.19, Pillai = 0.002, *p* = 7.83 × 10^−3^). These differences related almost entirely to increased levels of ASD gene expression in the pituitary gland (*F*_(1, 15317)_ = 14.10, p_corrected_ = 2.24 × 10^−3^; Figure 1). No other evaluated brain regions showed evidence of differential expression of ASD genes compared to genes in the random sets (Appendix A).

### 2.2. ASD Candidate Genes Associated with Mammalian Phenotypes

There were 225 biological processes defined in humans and overrepresented in ASD candidate genes. There were also 258 processes defined in mice that were overrepresented among ASD gene mouse orthologs; 200 of these overlapped with biological processes overrepresented for ASD genes in humans (Appendix A). Products from 931 genes (96.5% of all ASD genes) were involved in these processes. Seven terms reflecting Mammalian Phenotypes (MP) defined in the IMPC were mapped to a GO-defined biological process overrepresented among the ASD genes. Phenotypes included ‘abnormal postnatal growth/weight/body size’ (MP:0002089), ‘decreased body size’ (MP:0001265), ‘increased body size’ (MP:0001264), ‘abnormal nervous system morphology’ (MP:0003632), ‘abnormal brain development’ (MP:0000913), and ‘abnormal embryo development’ (MP:0001672). These traits were represented by three top level terms (i.e., ‘growth/size/body region phenotype’ [MP:0005378], ‘nervous system phenotype’ [MP:0003631], ‘embryo phenotype’ [MP:0005380]). More ASD genes (9.2%) were associated with at least one trait under these top-level terms in mouse knockouts when compared to genes in the random sets (Table 1, Χ^2^ = 5.42, p_corrected_ = 1.99 *×* 10^−2^). Specifically, more ASD genes were associated with abnormal postnatal growth/size/body region phenotypes when knocked out in mouse models (Table 2, Χ^2^ = 8.60, p_corrected_ = 1.01 *×* 10^−2^). The most prevalent of these types of traits was ‘decreased lean body mass’ (MP:0002089), followed by ‘decreased body length’ (MP:0002089; Figure 2).

### 2.3. ASD Candidate Genes Influencing Drug Response

Compared to random genes, ASD candidate genes encoded more potential drug targets or contain variants that may inform pharmaceutical treatment (Table 1). Specifically, an increased proportion of genes cited in connection with ASD (11.8%) encode FDA-approved drug targets (Χ^2^ = 219.18, p_corrected_ = 6.98 × 10^−49^), or had molecular properties similar to approved targets (15.5%, Χ^2^ = 60.25, p_corrected_ = 1.34 × 10^−14^). Furthermore, more ASD genes (13.0%) contained variants with evidence for significant pharmacogenetic effects (Χ^2^ = 103.25, p_corrected_ = 5.92 × 10^−24^).

### 2.4. ASD-ACMG Protein Interactions

There were nine ASD risk genes with evidenced pathogenic variants for which the ACMG recommends clinical testing (Appendix A). In addition, an increased proportion of proteins encoded by candidate genes (49.7%) had predicted direct interactions with proteins encoded by ACMG-recommended genes compared to proteins encoded by genes in the 1000 random sets (Table 1, Χ^2^ = 122.98, p_corrected_ = 3.76 × 10^−28^). On average, ASD-related proteins formed six connections (x = 6.32, sd = 14.20) with ACMG proteins; however, there was no evidence that candidate proteins formed more connections when compared to proteins encoded by genes in any of the random sets (Χ^2^ = 824.34, *p* = 9.99 × 10^−1^).

### 2.5. Evaluation of Pathogenic Variants in Prioritized ASD Candidate Genes

Of the 27,622 total genes harboring variants as curated in the September 2020 update of ClinVar (https://www.ncbi.nlm.nih.gov/clinvar/), 900 were genes cited in connection with ASD and 487 of these had a variant reported to be pathogenic or likely pathogenic. This represents approximately half (50.9%) of the full list of 956 ASD-related genes included in the DisGeNET. Of the 18 ASD-related genes that were prioritized via our pipeline (Figure 3), 78% (*n* = 14) had a variant reported to be pathogenic or likely pathogenic (Table 3, Appendix A).

## 3. Discussion

The central goal of this study was to build a bioinformatics pipeline to rapidly detect specific candidate genes that are more likely to be clinically relevant based on functional evidence and drug target properties. Compared to random gene sets, more ASD-related genes were expressed in control post-mortem brain tissue. In addition, more ASD genes were associated with postnatal growth phenotypes when knocked out in mice. Furthermore, ASD genes encoded more targets for FDA-approved drugs or bioactive molecules with drug-like properties, and more had a pharmacogenetic variant. Proteins encoded by ASD genes also showed more evidence for direct interactions with proteins encoded by the ACMG recommended clinically actionable genes. These results suggest that all of the functional categories incorporated in our pipeline were useful in identifying the candidate genes most likely to be functionally and clinically relevant. While there are numerous challenges limiting the clinical utility of information from genetic studies of complex conditions, the pipeline we developed addresses a key issue related to knowledge gaps among physicians regarding the benefits of genetic testing [44]. As incorporating genomic technology into patient care is largely dependent on clinicians’ perspectives of its utility, it is important to identify ways to better inform clinicians about relevant genetic findings beneficial to optimizing treatment. Developing tools that summarize the extensive amounts of data available and present them in a manner that offers more immediate insights into the benefits for patient care is a necessary step toward more efficient integration of this knowledge into the clinic. Ultimately, we expect evidence from our pipeline will be useful in supporting clinical decisions regarding the benefits of genetic testing for optimizing treatment. Results from our work may also inform future research focused on drug repurposing, characterizing pleiotropic genetic effects connecting clinically distinct conditions or prioritizing genes for in vitro or in vivo studies aimed at elucidating molecular mechanisms underlying expression of symptoms in ASD.

### 3.1. ASD-Related Gene Expression in the Pituitary Is Increased Compared to Random Sets

Although more ASD genes overall were expressed in the brain, evidence of differential expression was limited to the pituitary. Genes cited in connection with ASD were expressed at higher levels here compared to genes included in the random sets. During embryogenesis, non-dysfunctional genetic regulation of pituitary gland development results in typical production of numerous hormones and the appropriate release of neuropeptides, like oxytocin and arginine vasopressin [45]. Many of the hormones and neuropeptides produced and regulated by the pituitary have been suspected to be disrupted in individuals with ASD, as well as other neuropsychiatric disorders [46]. Notably, brain region specific gene expression data were obtained from individuals without evidence of disease [13]. It is possible that in individuals with ASD, dysregulation of these pituitary-expressed genes results in the development of a dysfunctional pituitary and subsequent issues with neurotransmission of molecules important to expression of social and repetitive behaviors, which are core symptoms of ASD [47]. It is unclear why other brain regions did not show evidence of differential expression of ASD candidate genes when compared to random sets of genes. While we chose to use the GTEx resource, as these data are more readily accessible—and likely more generalizable to other conditions—compared to resources containing data generated from ASD brain tissue, this may limit the ability to understand genetic differences that may be unique to the brains of individuals with ASD. Specifically, it is hypothesized that brain regions involved in social behavior drive ASD symptomatology; these include the amygdala, orbitofrontal cortex, temporoparietal cortex and insula [48]. With the exception of the amygdala, gene expression in the precise brain regions that are proposed biomarkers for ASD are not quantified in the GTEx resource potentially explaining why our results were limited to the pituitary. As opposed to prioritizing genes solely based on brain expression, focusing future work on prioritizing genes that are differentially expressed in the brains of patients with ASD may be more useful to pinpointing clinically relevant genetic data for ASD specifically.

### 3.2. ASD-Related Genes Associated with Abnormal Postnatal Growth in Mouse Knockouts

ASD candidates were also more likely to be associated with growth/size/body region phenotypes when knocked out in mouse models included in the Knockout Mouse Phenotyping (KOMP) project. This result supports a risk model for neurodevelopmental disorders that reflects deviations from typical body size during development [49]. However, it should be noted that while the ambitious goal of the KOMP project is to eventually test for associations between the extensive phenotypes included in the pipeline and every protein coding gene in the mouse, a number of genes have yet to be evaluated. For example, the *Phosphatase and Tensin Homolog* (*PTEN*) gene was excluded during this annotation step. *PTEN* is well-known to contain pathogenic clinically-actionable variants and has not yet been tested via the KOMP pipeline (https://www.mousephenotype.org/data/genes/MGI:109583). This is likely due to the fact that this gene is an essential gene and is homozygous lethal as a null resulting in a limited embryo phenotype [50]. Notably, *PTEN* has been well studied in disease-specific mouse KO projects and these data are available via the Mouse Genome Informatics (MGI) resource (http://www.informatics.jax.org/marker/key/31908). Unfortunately, MGI data are not readily accessible via an application program interface (API), like the IMPC data, making it more difficult to incorporate into an automated pipeline. Furthermore, there may be bias introduced when incorporating evidence obtained from data generated in disease-specific studies as many of these focus on confirming reports in humans and may only test for specific traits reflecting the implicated human disease [51,52]. It is expected that as the IMPC database expands to include more genes, the automated gene prioritization pipeline we developed will also improve. Future work may focus on determining the level of contribution mouse model data adds to the prioritization accuracy of our pipeline. In addition, we may specifically incorporate information regarding embryonic lethal phenotypes as part of the gene annotations.

### 3.3. More ASD Genes Are Implicated in Drug Response

Also notable is more ASD genes encode molecules that are useful to drug development, supporting the ability to maximize drug efficacy while minimizing adverse events. As mentioned above, numerous treatment regimens using pharmaceutical compounds have been developed to address specific symptoms and co-occurring conditions in ASD; however, there is insufficient evidence supporting efficacy for many compounds with many reported adverse events [7,8,9]. Functional characterization of implicated genes that encode targets for drugs not currently used to treat symptoms in ASD may offer opportunity for repurposing [53]. Additionally, identifying genes that encode similar molecules to current targets may help pinpoint genes and pathways that should be fast-tracked for functional study as novel targets. Furthermore, as many individuals carry pharmacogenetic variants that influence drug response [54], identifying these variants in ASD genes that encode targets for drugs used to treat ASD symptoms is an important avenue of research that may provide results helpful toward more effective personalized treatment of symptoms in ASD.

### 3.4. More ASD-Related Proteins Interact with ACMG Gene Encoded Proteins

Genes that are recommended for testing by the ACMG are almost entirely those containing variants having a strong relationship with increased cancer risk [42,55]. Given the mounting evidence that much of the genetic architecture underlying expression of cancer is shared with ASD [56], it is interesting that more ASD proteins were predicted to directly interact with ACMG proteins. It is possible that these results support a connection between ASD and cancer etiology. Regardless, identifying specific ASD-related proteins that interact with cancer susceptibility proteins may be important for recognizing genes with disease-causing variants that should be evaluated at the bench and in the clinic.

### 3.5. Prioritized ASD Candidate Genes More Likely to Have Pathogenic Variants

Ultimately, the pipeline we developed identified a subset of ASD-related genes with a higher proportion of pathogenic or likely pathogenic variants compared to the initial list of genes cited in connection with ASD. This suggests the approach is useful to deciphering clinically relevant genetic results from all of the currently available evidence. Notably, the pipeline is also useful in a broader sense for functional annotation of gene lists to select genes that are worthy of further study. By automating this process as much as possible, clinically useful results for hundreds of genes implicated in ASD can be delivered rapidly.

### 3.6. Limitations and Future Directions

Limitations for each step in the pipeline and future work aimed at overcoming these limitations have been noted throughout. Additional limitations of the approach include the inability to automate the pipeline in its entirety. This is due to the availability of many of the resources utilized. As detailed in the Material and Methods, resources that were considered important for ensuring accuracy of the pipeline (e.g., DRSC Integrative Ortholog Prediction Tool, Ontology Mapping repository) did not have an API thereby requiring direct downloads of these data. A future goal to circumvent automation issue is to establish a database with cached data to allow for complete automation. In addition, the genetic landscape is dynamic, meaning results obtained at one specific time point represent only a snapshot of the available information, and this may be potentially misleading. Future work will focus on regularly incorporating updates into pipeline automation to ensure the most current evidence is incorporated.

## 4. Materials and Methods

A schematic of the entire functional annotation pipeline developed is provided in Figure 4. More specifically, an initial list of ASD candidate genes was identified by querying data available in the DisGeNET 6.0 database of human gene-disease associations [17], v1.1.0 update May 2019 (http://www.disgenet.org). The benefits of this database include gene-disease associations identified via text-mining of multiple sources and includes evidence from studies of idiopathic ASD cases (including cases from multiplex families and sporadic cases from simplex families), as well as studies conducted in individuals with underlying familial and non-familial syndromes. All Unified Medical Language System Concept Unique Identifiers (CUI) that relate to ASD were selected from the disease mappings file provided by DisGeNET and a data frame of genes with evidence for a relationship with CUIs relating to ASD was created. CUIs queried in DisGeNET were as follows: Autism Spectrum Disorders [C1510586], Autistic behavior [C0856975], and Autistic Disorder [C0004352], Autistic features [C1846135], Autistic spectrum disorder with isolated skills [C1298684].

Next, the list of ‘ASD-related’ genes were annotated to identify those expressed in the human brain. We used RNA-sequencing data from non-diseased brain tissue made available via the Genotype-Tissue Expression (GTEx) project, an ongoing collaborative effort to build a comprehensive public resource to study tissue-specific gene expression and regulation [13,57]. Genes expressed at a default cut-off of ≥ 0.5 transcripts per million (TPM) across all available brain regions in GTEx were downloaded from the Expression Atlas (https://www.ebi.ac.uk/gxa/home) and queried for ASD-related genes. Available brain regions included the amygdala (*n* = 129 individuals), anterior cingulate cortex (*n* = 147), caudate (*n* = 194), cerebellar hemisphere (*n* = 175), cerebellum (*n* = 209), frontal cortex (*n* = 175), hippocampus (*n* = 165), hypothalamus (*n* = 170), nucleus accumbens (*n* = 202), putamen (*n* = 170) and substantia nigra (*n* = 114).

To further identify genes encoding proteins that function in biological processes relevant to ASD etiology, conditional gene set overrepresentation analyses were conducted comparing candidate genes to all human protein coding genes included in Ensembl release 99 [58]. We used the parent-child algorithm from the TopGO R package version 2.38.1 (The R Foundation for Statistical Computing, Vienna, Austria) for overrepresentation analyses which incorporates knowledge about hierarchical relationships between GO terms into the calculation of statistical significance [59,60]. Significance was determined using Fisher’s exact test and set at a Bonferroni-adjusted level of α = 0.05/*n* where *n* is the number of nodes tested. As a goal was to find genes with a phenotypic consequence when knocked out in mice, we then mapped human genes to the most likely mouse orthologs using the best match from all prediction tools available via the DRSC Integrative Ortholog Prediction Tool (Version 8.0 August 2019; https://www.flyrnai.org/diopt). Only orthologs that had a high rank, indicating that the number of tools that support the orthologous gene-pair relationship was ≥ 2, and that this ortholog had the best score in both forward and reverse mapping, were included in subsequent analyses [61]. Overrepresentation analyses were then conducted comparing mouse orthologs for ASD genes to all mouse protein coding genes as described above for human genes. Biological processes overrepresented for human ASD genes and mouse ASD gene orthologs were cross-referenced to identify overlap. We then mapped significantly overrepresented GO terms defined in both humans and mice to mammalian phenotype terms (MP) using direct mapping (i.e., distance = 1) from the Ontology Mapping repository, OxO (https://www.ebi.ac.uk/spot/oxo, updated 11 September 2020), which is hosted by the Ontology Lookup Service [62]. The genotype-phenotype representational state transfer API from the IMPC (http://www.mousephenotype.org/) [63] was subsequently queried using the jsonlite R package v1.6.1 (The R Foundation for Statistical Computing, Vienna, Austria) [64] to prioritize genes that have a phenotypic consequence—associated with *p* ≤ 0.05 in KOs of either sex—representing top level MP terms that reflect the ASD gene overrepresented GO-defined processes.

To identify genes encoding currently approved drug targets or those that may be novel drug targets, we used jsonlite to query the Pharos database (https://pharos.nih.gov/idg/api). Available drug development levels included: FDA-approved compound targets (Tclin), molecules with known properties similar to approved drug targets (Tchem), proteins with known biological or molecular functions but no known drug target properties (Tbio), and proteins with relatively unknown function (Tdark). To identify genes with pharmacogenetic variants evidenced to influence individual responses to drugs, we directly downloaded variant and clinical annotations data from PharmGKB (https://www.pharmgkb.org/downloads), update 20 March 2020. Only those associations that were reported in the literature as significant were used in annotations.

To decipher genes that are potentially actionable, protein-protein interactions between ASD candidate proteins and proteins currently recommend as clinically actionable by the ACMG were predicted using STRINGdb R package v10 (The R Foundation for Statistical Computing, Vienna, Austria) [65]. The ACMG 2016 update SF v2.0 [42] was used for network prediction (https://www.ncbi.nlm.nih.gov/clinvar/docs/acmg/). We focused on ASD-related proteins with evidence for direct interactions with proteins encoded by ACMG-recommended genes. We also calculated the quantity of direct interactions between proteins in this network. ACMG-ASD protein interaction networks were visualized using Cytoscape v3.5.1 (The Cytoscape Consortium, San Diego, CA, USA) [66].

To help show that the pipeline was useful for prioritizing clinically relevant candidate genes, we selected 1000 random sets of protein coding genes in humans (Ensembl release 99), while allowing for replacement, of equal number to the ASD gene list from biomaRt R package v 2.42.1 (The R Foundation for Statistical Computing, Vienna, Austria) [67] and annotated as described above. The one proportion test was used to determine if the proportion of genes in the ASD set that were annotated was increased compared to the average proportions of genes across all random sets (i.e., the expected proportion). Significance was based on a Benjamin-Hochberg false discovery rate corrected at *p* ≤ 0.05. To determine if specific brain regions were enriched for ASD candidate gene expression compared to random genes, we used the limma package in R v 3.38.3 (The R Foundation for Statistical Computing, Vienna, Austria) [68]. The average expression of each gene set in the respective brain regions was clustered based on Euclidean measures. Tests for differences in the average gene expression of ASD genes and the average expression across genes in the random sets among brain regions were done using multivariate analysis of variance. Significance was based on a Benjamini-Hochberg false discovery rate corrected at *p* < 0.05. Kruskal-Wallis test was used to determine if the number of direct connections between ACMG-recommended gene encoded proteins were different across the ASD and random sets. Evidence included in ClinVar expanding upon current ACMG recommendations was also used as a gold standard reference to determine how often ASD genes with confirmed pathogenic or likely pathogenic variants were prioritized.

## Figures and Tables

**Figure 1 ijms-21-09029-f001:**
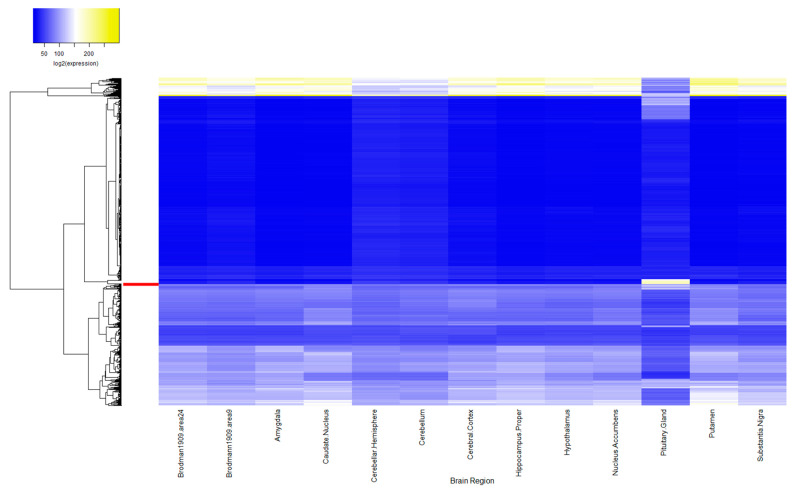
Brain Expression Profiles of ASD Candidate Gene Set and Random Gene Sets. Shown are average expression levels for ASD and random gene sets, based on transcripts per million, for each brain region obtained from typical human tissue in GTEx. Average expression of each gene set in the respective brain region are clustered based on Euclidean measures. The ASD risk gene cluster has been amplified for easier visualization and is highlighted in red on the y-axis.

**Figure 2 ijms-21-09029-f002:**
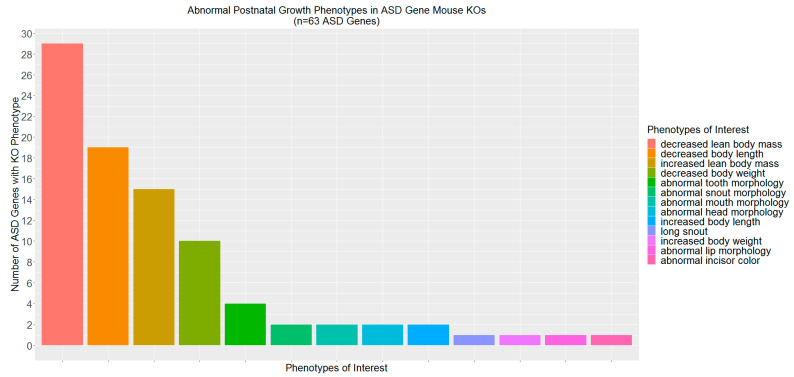
Frequency of Autism Spectrum Disorder-related Genes Associated with Abnormal Postnatal Growth Phenotypes. Shown are the specific traits that were associated more often with knocking out the 63 ASD candidate genes associated with growth phenotypes in mice (see Appendix A for mouse trait associated genes) when compared to genes from random sets. Traits are presented in descending order based on the number of associated ASD genes.

**Figure 3 ijms-21-09029-f003:**
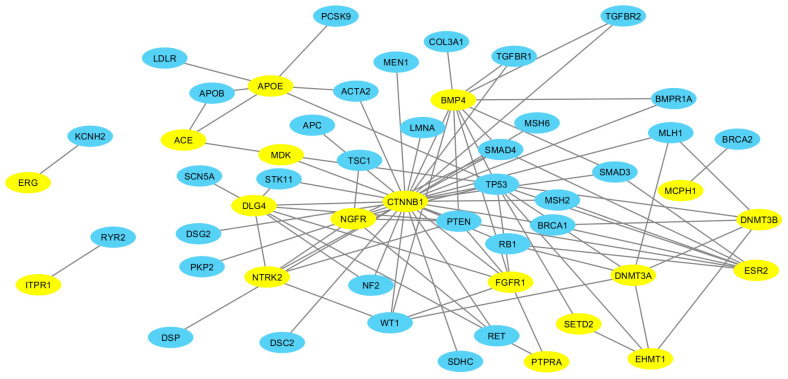
Prioritized ASD Proteins and ACMG Proteins Interaction Network. Shown are direct interactions predicted between proteins encoded by ASD candidate genes annotated in all pipeline categories and proteins encoded by American College of Medical Genetics (ACMG) recommended actionable genes. Top ASD-related proteins are highlighted in yellow, ACMG-recommended proteins in blue. Notably, none of the prioritized ASD genes were represented among the 59 genes compiled by the ACMG.

**Figure 4 ijms-21-09029-f004:**
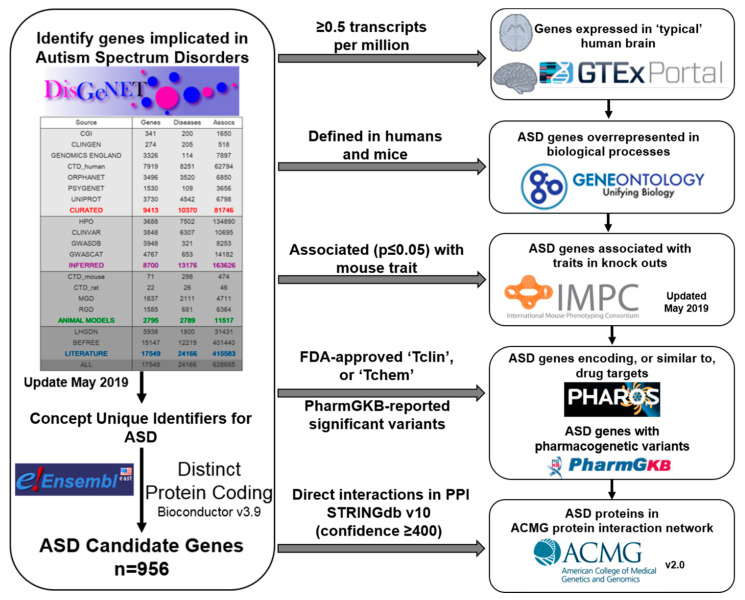
Candidate Gene Identification and Annotation Pipeline. All human protein coding genes defined in Ensembl, with evidence for influencing risk for concept unique identifiers reflecting Autism Spectrum Disorder (ASD) were identified from the DisGeNET resource which compiles evidence from all sources included in the corresponding table on the left side of the schematic. The initial list of 956 ASD candidate genes was then annotated using the publicly available resources included on the right side of the schematic, based on the criteria indicated above each arrow for each annotation step. TPM = transcripts per million, Tclin = FDA-approved compound targets, Tchem = molecules with properties similar to approved targets.

**Table 1 ijms-21-09029-t001:** Comparisons of Functional Annotations for Autism Spectrum Disorder Candidate Genes to Random Genes.

Gene Attribute	ASD (*n*)	Random (mean ± sd)	Χ^2^ (95%CI)	*p*-Value	*p*-Value_corrected_
Brain Expressed	861	808.94 ± 11.01	21.36 (0.88, 0.92)	3.80 × 10^−6^	5.07 × 10^−6^
Associated Mouse Trait	88	68.89 ± 7.83	5.42 (0.07, 0.11)	1.99 × 10^−2^	1.99 × 10^−2^
Encodes Tclin	113	31.19 ± 5.45	219.18 (0.10, 0.14)	1.37 × 10^−49^	6.98 × 10^−49^
Encodes Tchem	148	80.76 ± 8.43	60.25 (0.13, 0.18)	8.35 × 10^−15^	1.34 × 10^−14^
Encodes Tbio	615	574.56 ± 14.99	6.96 (0.61, 0.67)	8.34 × 10^−3^	9.53 × 10^−3^
Encodes Tdark	51	253.28 ± 13.02	218.69 (0.04, 0.07)	1.74 × 10^−49^	6.98 × 10^−49^
Pharmacogenomic	124	52.15 ± 50.23	103.25 (0.11, 0.15)	2.96 × 10^−24^	5.92 × 10^−24^
ACMG Network	475	313.53 ± 13.25	122.98 (0.46, 0.53)	1.41 × 10^−28^	3.76 × 10^−28^
All Attributes	18	8.02 ± 2.56	11.30 (0.01, 0.03)	7.75 × 10^−4^	-

Shown are the number of genes cited in connection with ASD in DisGeNET with pipeline attributes compared to the average number of genes across 1000 random sets. Included for each attribute are results of chi-square tests comparing the proportion of genes annotated in the ASD list (*n* = 956) to the average proportion of genes across all random lists. Uncorrected and Benjamini-Hochberg corrected *p*-values are included. Tclin = FDA-approved compound targets, Tchem = molecules with known properties similar to approved drug targets, Tbio = proteins with known biological or molecular functions but no known drug target properties, Tdark = proteins with relatively unknown function, sd = standard deviation, CI = confidence interval.

**Table 2 ijms-21-09029-t002:** Proportions of Autism Spectrum Disorder Genes Associated with Specific Top Level Mammalian Phenotypes.

Associated Mouse Trait	ASD (*n*)	Random (mean ± sd)	Χ^2^ (95%CI)	*p*-Value	*p*-Value_corrected_
Growth phenotype	63	43.59 ± 6.32	8.60 (0.05, 0.08)	3.37 × 10^−3^	1.01 × 10^−2^
Nervous system phenotype	18	12.16 ± 3.36	2.37 (0.01, 0.03)	1.23 × 10^−1^	1.85 × 10^−1^
Embryo phenotype	17	18.46 ± 4.18	0.05 (0.01, 0.03)	8.21 × 10^−1^	8.21 × 10^−1^

Shown are results comparing the number of ASD candidate genes that were associated with a mammalian phenotype reflecting overrepresented Gene Ontology Biological Processes defined in humans, compared to the average number of genes across 1000 random sets. Chi-square test results are based on the proportion of genes in the ASD list (*n* = 956) compared to the average proportion of genes across all random lists. Uncorrected and Benjamini-Hochberg corrected *p*-values are included. sd = standard deviation.

**Table 3 ijms-21-09029-t003:** Prioritized Genes with Pathogenic/Likely Pathogenic Variants.

ASD Gene	Brain Region (TPM)	Associated Mouse Trait (s)	Mapped GO BP	Drug Development	PharmGKB
*APOE*	Substantia Nigra (1141)	postnatal growth	growth	NA	PA55
*DLG4*	Cerebellar Hemisphere (232)	postnatal growth	growth	Tchem	NA
*CTNNB1*	Cerebellar Hemisphere (188)	postnatal growth	growth	Tchem	PA27013
*FGFR1*	Cerebellum (92)	embryo development	organism development	Tclin	NA
*NTRK2*	Brodman1909 area24 (91)	postnatal growth	growth	Tchem	PA31818
		nervous system morphology	CNS development		
*ITPR1*	Cerebellum (73)	embryo development	organism development	Tchem	NA
*SETD2*	Cerebellar Hemisphere (41)	postnatal growth	growth	Tchem	NA
		nervous system morphology	CNS development		
*DNMT3A*	Cerebellum (21)	postnatal growth	growth	Tclin	PA27445
		embryo development	organism development		
*EHMT1*	Cerebellum (14)	postnatal growth	growth	Tchem	NA
*BMP4*	Cerebellar Hemisphere (8)	nervous system morphology	CNS development	Tchem	NA
*ACE*	Pituitary (7)	embryo development	organism development	Tclin	PA139
*DNMT3B*	Cerebellar Hemisphere (6)	embryo development	organism development	Tchem	NA
*MCPH1*	Cerebellar Hemisphere (6)	postnatal growth	growth	NA	PA30701
*ESR2*	Pituitary (0.8)	postnatal growth	growth	Tclin	PA27886

Details are shown for the 14 Autism Spectrum Disorder (ASD) genes annotated in all functional categories included in the pipeline that had pathogenic variants defined in ClinVar. Included are human gene symbols, the brain region of highest expression in Genotype Tissue Expression (GTEx), traits associated in mouse knockouts, the corresponding Gene Ontology (GO) Biological Process (BP) defined in humans, the drug development level, and the PharmGKB ID for pharmacogenetic variant guidelines. TPM = transcripts per million, Tclin = FDA-approved compound targets, Tchem = molecules with known properties similar to approved targets, CNS = central nervous system, NA = not available.

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
