# Peer review of "An Automated Functional Annotation Pipeline That Rapidly Prioritizes Clinically Relevant Genes for Autism Spectrum Disorder"

_ijms, 2020, doi:10.3390/ijms21239029_

Round 1

Reviewer 1 Report

The authors have addressed my concerns accordingly. Therefore, I decided to accept this manuscript.

Reviewer 2 Report

In this work, the authors present an automated Functional annotation pipeline that rapidly prioritizes clinically relevant genes for autism spectrum disorder

The topic is interesting, and it tackles a question dealt in the literature using an innovative approach. It has practical applications.
The description of goals is clear, as well as the methodology applied.

The overall result of the manuscript is very promising.

I believe that the paper is well written and organised and for these reasons I suggest to accept it after minor revisions listed below:

-the authors should add a figure that summarizes the pipeline;

-the authors should improve the quality of the figures.

Author Response

This manuscript is a resubmission of an earlier submission. The following is a list of the peer review reports and author responses from that submission.

Round 1

Reviewer 1 Report

Veatch et al., performed computational analyses using a first set of (potential) ASD-related genes (N=956) to pinpoint those with likely clinical relevance based on other evidence such expression in brain, phenotypes in knock-out genes in mice, genes in ACMG, drug response, etc. The manuscript is very well written. However, the workflow looks to me more an excursus of random enrichment analyses that have little to do with the main aim or have little connection with each other. Most of the statistical tests can be performed with a more appropriate hypergeometric test. From 956 genes related to ASD in DisGeNET, only 18 are found in all categories. The objective of this study is finding clinically relevant variants and genes. Unfortunately, the number of genes implicated in ASD is extremely large and this pipeline does not allow to narrow down a number of them that may be used for clinical purposes. Most of the genes excluded are actually ASD genes, and the workflow used here disregarded their role offering 18 genes that have no reason to be considered better ASD candidate genes than other. It is not possible base clinical strategies on a such limited number of genes, which together would explain a very small number of cases. The discussion is not focused on results and diluted with other weak connections/considerations (i.e. cancer and PTEN)

Reviewer 2 Report

This manuscript is exposing the important software parameters from the research accordingly, so it will possibly facilitate easier replication of the pipeline run by the other groups.

There are some minor issues that should be clarified though:

Introduction section:

Line 42-49: It would be good for the reader if more insight why ASD should be examined with more information on its prevalence (1 in 160 children) and existing available intervention. The link is here: https://www.who.int/news-room/fact-sheets/detail/autism-spectrum-disorders
The insufficiency of existing intervention will be a good starting point to narrate why precision medicine is important for ASD studies. 

Results section:

Line 104: I see that you found 956 genes with association to ASD, and this is a basis for your benchmarking effort to the 1000 random gene set. Why use 1000 random gene set instead of 956?

Discussion section:

Line 229: Why the differential expression was limited mainly to the pituitary gland? Any specific explanation? Is there any biomarkers available others than from the pituitary?

Limitations and Future Directions section:

Line 229-306: It should be more specific why those resources could not be incorporated into the pipeline. Why those resources, that explained as 'many' could be a hindrance to automate the whole pipeline? Is there any computational power (time and/or space complexity) constraint?

Round 2

Reviewer 1 Report

Unfortunately, my previous concerns for this study have not been resolved. The authors focus their responses on ASD genes implicated in syndromic condition: ‘We defined clinical relevance in this project as any ASD associated gene with evidence indicating a patient may have a co-occurring or underlying syndromic condition that requires direct intervention, or is relevant to drug development and/or approaches to maximizing efficacy and minimizing adverse events” and redundant sentences. I want to clarify that genetic studies in ASD probands are performed in idiopathic cases, and Syndromic cases of ASD follow mendelian genetic approaches. In summary, the large genetic studies are interested to map genes implicated in idiopathic ASD condition.